# An autonomous laboratory for the accelerated synthesis of inorganic materials

Nathan J. Szymanski[1,2,5], Bernardus Rendy[1,2,5], Yuxing Fei[1,2,5], Rishi E. Kumar[3,5], Tanjin He[1,2], David Milsted[2], Matthew J. McDermott[1,2], Max Gallant[1,2], Ekin Dogus Cubuk[4], Amil Merchant[4], Haegyeom Kim[2], Anubhav Jain[3], Christopher J. Bartel[2], Kristin Persson[1,2], Yan Zeng[2 ✉] & Gerbrand Ceder[1,2 ✉]

To close the gap between the rates of computational screening and experimental realization of novel materials[1,2], we introduce the A-Lab, an autonomous laboratory for the solid-state synthesis of inorganic powders. This platform uses computations, historical data from the literature, machine learning (ML) and active learning to plan and interpret the outcomes of experiments performed using robotics. Over 17 days of continuous operation, the A-Lab realized 36 compounds from a set of 57 targets including a variety of oxides and phosphates that were identified using large-scale ab initio phase-stability data from the Materials Project and Google DeepMind. Synthesis recipes were proposed by natural-language models trained on the literature and optimized using an active-learning approach grounded in thermodynamics. Analysis of the failed syntheses provides direct and actionable suggestions to improve current techniques for materials screening and synthesis design. The high success rate demonstrates the effectiveness of artificial-intelligence-driven platforms for autonomous materials synthesis and motivates further integration of computations, historical knowledge and robotics.

Although promising new materials can be identified at scale using high-throughput computations, their experimental realization is often challenging and time-consuming. Accelerating the experimental segment of materials discovery requires not only automation but autonomy—the ability of an experimental agent to interpret data and make decisions based on it. Pioneering efforts have demonstrated autonomy in several aspects of materials research, including robotic and Bayesian-driven optimization of carbon nanotube yield[3,4], photovoltaic performance[5] and photocatalysis activity[6]. In contrast to conventional ML algorithms used for optimization, human researchers benefit from a wealth of background knowledge that informs their decision-making, and it is increasingly recognized[7–9] that autonomy will require a fusion of encoded domain knowledge, access to diverse data sources and active learning.

Here we present the A-Lab, an autonomous laboratory that integrates robotics with the use of ab initio databases, ML-driven data interpretation, synthesis heuristics learned from text-mined literature data and active learning to optimize the synthesis of inorganic materials in powder form. Although autonomous workflows based on liquid handling have been demonstrated in organic chemistry[10–13], the A-Lab addresses the unique challenges of handling and characterizing solid inorganic powders. These often require milling to ensure good reactivity between precursors, which can have a wide range of physical properties related to differences in their density, flow behaviour, particle size, hardness and compressibility. The use of solid powders is well suited for manufacturing and technological scaleup, and the approach of the A-Lab to synthesis produces multigram sample quantities that facilitate device-level testing.

Given a set of air-stable target materials (that is, desired synthesis products whose yield we aim to maximize) screened using the Materials Project[14], the A-Lab generates synthesis recipes using ML models trained on historical data from the literature and then performs these recipes with robotics. The synthesis products are characterized by X-ray diffraction (XRD), with two ML models working together to analyse their patterns. When synthesis recipes fail to produce a high target yield, active learning closes the loop by proposing improved follow-up recipes. Over 17 days of operation, the A-Lab successfully synthesized 36 of 57 target materials that span 33 elements and 40 structural prototypes (Supplementary Fig. 1 and Supplementary Table 1). Inspection of the 17 unobtained targets revealed synthetic as well as computational failure modes, several of which could be overcome through minor adjustments to the lab's decision-making. With its high success rate in validating predicted materials, the A-Lab showcases the collective power of ab initio computations, ML algorithms, accumulated historical knowledge and automation in experimental research.

## Autonomous materials-synthesis platform

The materials synthesis and characterization procedures used in the A-Lab are shown in Fig. 1. All target materials considered in this work are new to the lab, that is, not present in the training data for the algorithms it uses to propose synthesis recipes (Methods). The experiments

[1]Department of Materials Science and Engineering, University of California, Berkeley, Berkeley, CA, USA. [2]Materials Sciences Division, Lawrence Berkeley National Laboratory, Berkeley, CA, USA. [3]Energy Technologies Area, Lawrence Berkeley National Laboratory, Berkeley, CA, USA. [4]Google DeepMind, London, UK. [5]These authors contributed equally: Nathan J. Szymanski, Bernardus Rendy, Yuxing Fei, Rishi E. Kumar. ✉e-mail: yanzeng@lbl.gov; gceder@berkeley.edu

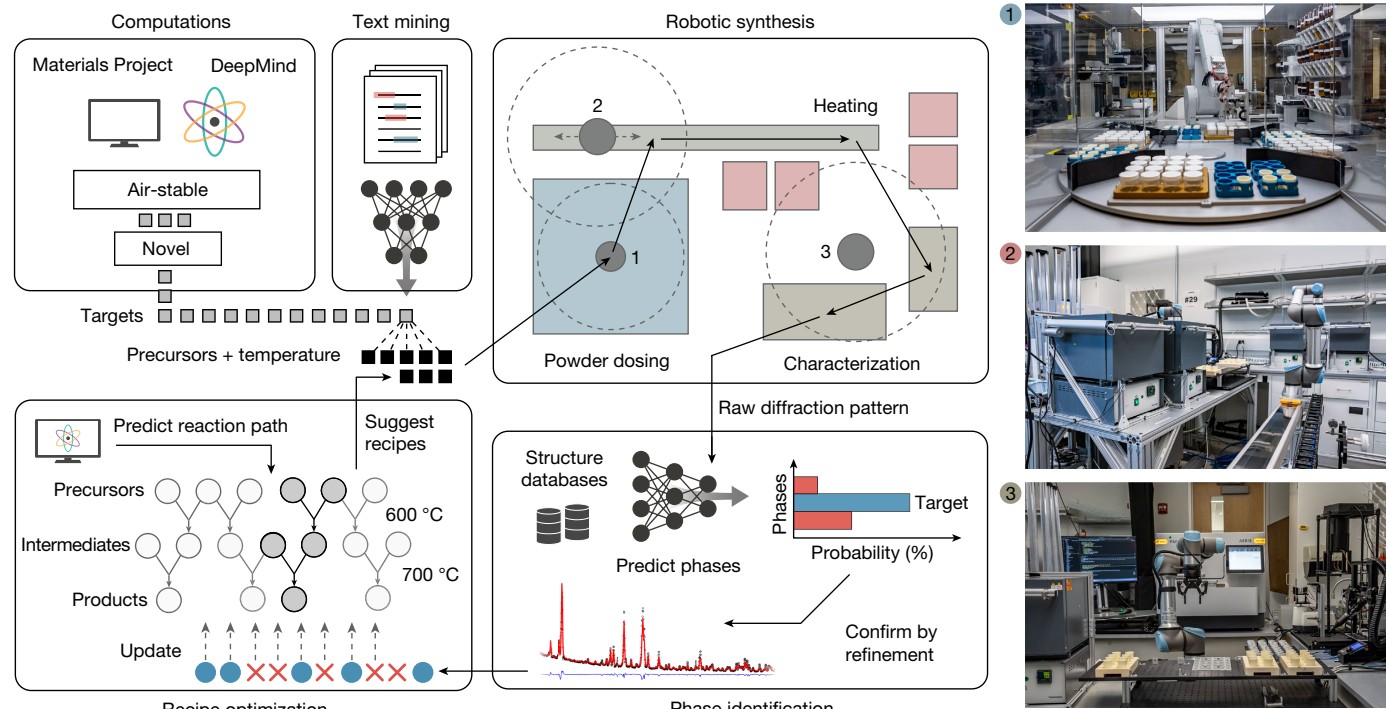

**Computations** — Materials Project, DeepMind, Air-stable, Novel, Targets

**Text mining**

**Robotic synthesis** — Heating, Powder dosing, Characterization

Precursors + temperature

Predict reaction path — Suggest recipes — Precursors, Intermediates, Products, 600 °C, 700 °C — Update — Recipe optimization

Structure databases — Predict phases — Raw diffraction pattern — Phases, Target, Probability (%) — Confirm by refinement — Phase identification

**Fig. 1 | Autonomous materials synthesis with the A-Lab.** Air-stable targets are identified using DFT-calculated convex hulls consisting of ordered ground states from the Materials Project and Google DeepMind. Synthesis recipes for each target are proposed using ML models trained on synthesis data from the literature. These recipes are tested using a robotic laboratory that automates (1) powder dosing, (2) sample heating and (3) product characterization with XRD. All sample transfer between these stations is performed using robotic arms, forming a fully automated sequence from chemical input to characterization. Phase purity is assessed from XRD, which is analysed by ML models trained on structures from the Materials Project and the ICSD, and confirmed with automated Rietveld refinement. In cases in which high (>50%) target yield is not obtained, new synthesis recipes are proposed by an active-learning algorithm that identifies reaction pathways with maximal driving force to form the target.

reported in this study represent the first attempts by the A-Lab to synthesize any of these targets. Each target is predicted to be on or very near (<10 meV per atom) the convex hull formed by stable phases taken from the Materials Project[14] and cross-referenced with an analogous database from Google DeepMind. Because the A-Lab handles samples in open air, we only considered targets that are predicted not to react with $O_2$, $CO_2$ and $H_2O$ (Methods). Like all databases comprised of atomistic calculations[15–17], the Materials Project represents its compounds as ordered structures, even in cases where these materials may form in experiments with partial disorder. This approach is commonly used in theory-driven materials screening efforts as ab-initio computations cannot represent partial atoms[18,19]. As such, we consider a synthesis procedure successful when it forms either an ordered or partially disordered version of its target material[20,21].

For each compound proposed to the A-Lab, up to five initial synthesis recipes are generated by a ML model that has learned to assess target 'similarity' through natural-language processing of a large database of syntheses extracted from the literature[22], mimicking the approach of a human to base an initial synthesis attempt on analogy to known related materials. A synthesis temperature is then proposed by a second ML model trained on heating data from the literature[23] (Methods). If these literature-inspired recipes fail to produce >50% yield for their desired targets, the A-Lab continues to experiment using Autonomous Reaction Route Optimization with Solid-State Synthesis (ARROWS[3]), an active-learning algorithm that integrates ab initio computed reaction energies with observed synthesis outcomes to predict solid-state reaction pathways[24]. Experiments are performed under the guidance of this algorithm until the target is obtained as the majority phase or all synthesis recipes available to the A-Lab are exhausted.

The A-Lab carries out experiments using three integrated stations for sample preparation, heating and characterization, with robotic arms transferring samples and labware between them (Fig. 1 and Extended Data Figs. 1 and 2). The first station dispenses and mixes precursor powders before transferring them into alumina crucibles. A robotic arm from the second station loads these crucibles into one of four available box furnaces to be heated (Methods). After allowing the samples to cool, another robotic arm transfers them to the third station, where they are ground into a fine powder and measured by XRD. The operations of the lab are controlled through an application programming interface, which enables on-the-fly job submission from human researchers or decision-making agents (Extended Data Fig. 3).

The phase and weight fractions of the synthesis products are extracted from their XRD patterns by probabilistic ML models trained on experimental structures from the Inorganic Crystal Structure Database (ICSD) following the methodology outlined in previous work[25,26]. Because the A-Lab is given no prior experimental reports for the target materials considered in this work, their diffraction patterns are simulated from computed structures available in the Materials Project and corrected to reduce density functional theory (DFT) errors (Supplementary Note 1). For each sample, the phases identified by ML are confirmed with automated Rietveld refinement (Methods and Supplementary Note 2) and the resulting weight fractions are reported to the management server of the A-Lab to inform subsequent experimental iterations, if necessary, in search of an optimal recipe with high target yield.

## Experimental synthesis outcomes

Using the described workflow, the A-Lab synthesized 36 of the 57 target compounds over 17 days of continuous experimentation, representing a 63% success rate. To confirm whether the A-Lab successfully made these targets, all diffraction patterns were later manually refined and

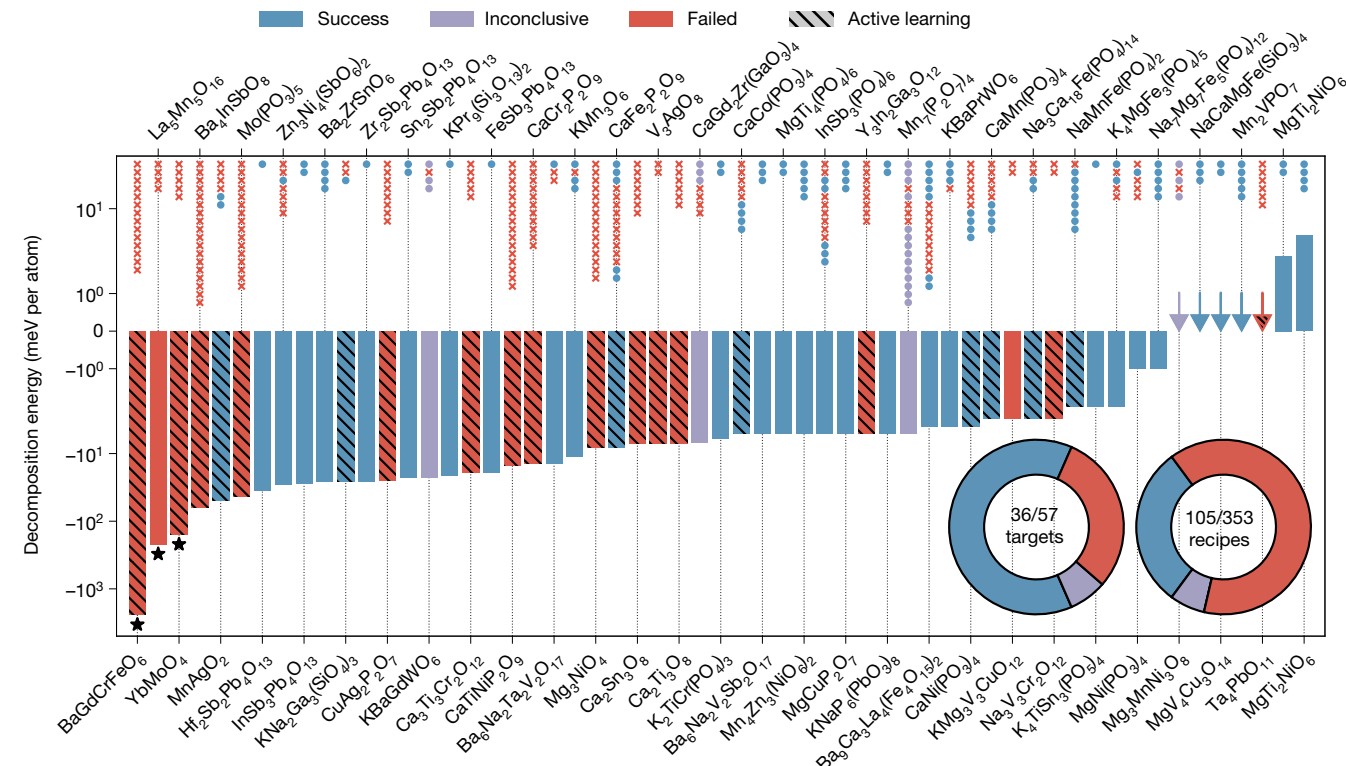

**Fig. 2 | Outcomes from targeted syntheses of 57 inorganic materials.**
Results summary from the synthesis efforts targeting 57 compounds, plotted against their decomposition energies (log scale). Arrows indicate values near zero. A total of 36 targets were successfully synthesized (blue bars), four were inconclusive from XRD (purple bars), and the remaining 17 were not obtained by the A-Lab (red bars). Targets optimized in the active-learning stage of the A-Lab are marked by diagonal lines; all other targets were only attempted using recipes proposed by ML algorithms trained on literature data. The scatter points above each bar represent the outcomes of attempted recipes for each target, ordered from top to bottom in the sequence in which they were performed. The inset pie charts show the fraction of successful targets (left) and recipes (right). Analyses performed after the fact suggest that the calculated decomposition energies for three targets, marked with stars, may be suspect owing to computational errors (see text).

compared with the A-Lab's autonomously made refinements and conclusions (Supplementary Data). Our manual analysis confirms that in each of the 36 successes, a satisfactory fit to the experimental diffraction pattern can be obtained only when the target is included in refinement. This does not necessarily imply the target was made with high purity, as many of the samples contain prominent byproducts that likely result from incomplete reactions. The A-Lab used a short hold time of 4 h during its syntheses, which could be extended and combined with intermittent regrinding to improve the phase purity. We also note that while the automated XRD interpretation may not identify multi-phase mixtures conclusively, it is generally reliable in identifying the majority phase, which is then used to guide synthesis optimization. We show in the next section that this success rate could be improved to 67% with only minor modifications to the lab's decision-making algorithm, and further to 70% if the computational techniques were also improved. Additionally, there were four materials (totaling 40/57 successes) whose presence in the synthesis products was originally reported by the A-Lab, but manual reexamination deems inconclusive from XRD alone due to ambiguity between the target's diffraction pattern and other known phases.

The high success rate demonstrates that comprehensive ab initio calculations can be used to effectively identify synthesizable materials. The outcome for all 57 compounds is plotted in Fig. 2 against their decomposition energies (on a log scale), a common thermodynamic metric that describes the driving force to form a compound from its neighbours on the phase diagram[27] (Supplementary Fig. 2). A negative (positive) decomposition energy indicates that a material is stable (metastable) at 0 K. Of the targets considered in this work, 50 are predicted to be stable, whereas the remaining seven are metastable but lie near

the convex hull. Over the range of decomposition energies considered, we do not observe a clear correlation between decomposition energy and whether a material was successfully synthesized.

In total, 30 of the 36 materials synthesized by the A-Lab were obtained using recipes proposed by ML models trained on synthesis data from the literature (Supplementary Note 3). These literature-inspired recipes were more likely to succeed when the reference materials are highly similar to our targets (Supplementary Fig. 3), confirming that target 'similarity' is a useful metric to select effective precursors[28]. At the same time, precursor selection remains a highly nontrivial task, even for thermodynamically stable materials. Despite 63% of targets eventually being obtained, only 30% of the 353 synthesis recipes tested by the A-Lab produced their targets. This finding echoes previous work that has established the strong influence of precursor selection on the synthesis path, ultimately deciding whether it forms the target or becomes trapped in a metastable state[29-32].

The active-learning cycle of the A-Lab[24] identified synthesis routes with improved yield for nine targets, of which six had zero yield from the initial literature-inspired recipes. Targets optimized with active learning are indicated by the bars containing diagonal lines in Fig. 2. In this framework, improved synthesis routes are designed using two hypotheses: (1) solid-state reactions tend to occur between two phases at a time (that is, pairwise)[33-35] and (2) intermediate phases that leave only a small driving force to form the target material should be avoided, as they often require long reaction time and high temperature[29,30,36].

The A-Lab continuously builds a database of pairwise reactions observed in its experiments—88 unique pairwise reactions (Supplementary Table 2) were identified from the synthesis experiments performed in this work. This database allows the products of some recipes to be

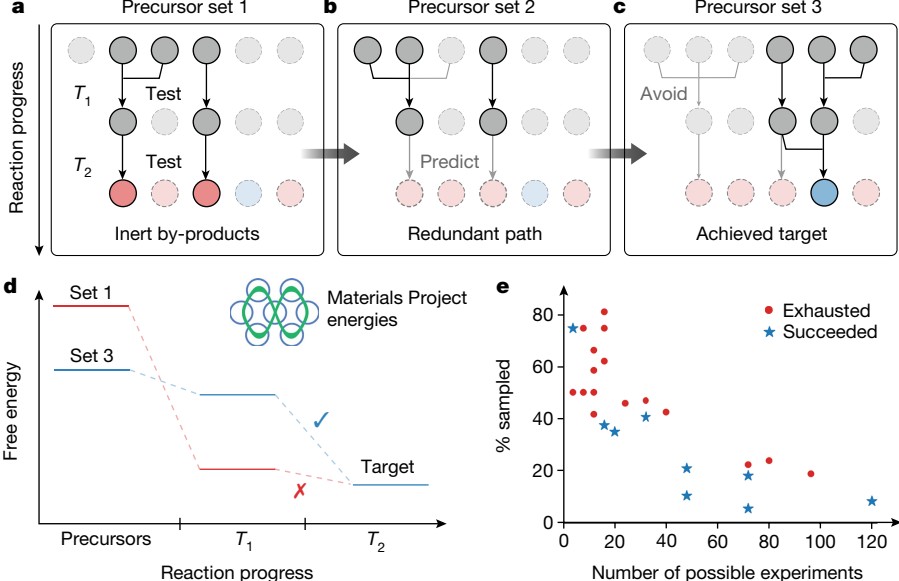

**Fig. 3 | Active learning with pairwise reaction analysis. a**, From a failed synthesis attempt, the A-Lab determines which pairwise reactions occurred. **b**, New precursors are recommended by substituting at least one precursor involved in the unfavourable pairwise reaction. In cases in which the new precursor set leads to identical intermediates as a previously tested set, it is not explored at any higher temperatures. **c**, The successful precursor set avoids all the unfavourable pairwise reactions. **d**, The free energy at each step

in the reaction pathway, calculated using data from the Materials Project, which shows that the successful pathway maintains a large driving force at the target-forming step. **e**, Following this approach, the scatter plot shows the number of experiments required to exhaust all unique reaction paths for each target (red) or to identify an optimal path with high yield (blue), plotted with respect to the total size of each experimental search space.

inferred, precluding their testing; a recipe that yields an observed set of intermediates (already present in the lab's database) need not be pursued at higher temperatures, as the remaining reaction pathway is already known (Fig. 3a,b). This can reduce the search space of possible synthesis recipes by up to 80% when many precursor sets react to form the same intermediates (Fig. 3e and Supplementary Notes 4 and 5). Furthermore, knowledge of reaction pathways can be used to give priority to intermediates with a large driving force to form the target, computed using formation energies available in the Materials Project (Fig. 3c,d). For example, the synthesis of $CaFe_2P_2O_9$ was optimized by avoiding the formation of $FePO_4$ and $Ca_3(PO_4)_2$, which have a small driving force (8 meV per atom) to form the target. This led to the identification of an alternative synthesis route that forms $CaFe_3P_3O_{13}$ as an intermediate, from which there remains a much larger driving force (77 meV per atom) to react with CaO and form $CaFe_2P_2O_9$, causing an approximately 70% increase in the yield of the target (Supplementary Note 6).

## Barriers to synthesis

Seventeen of the 57 targets evaluated by the A-Lab were not obtained even after its active-learning cycle. We identify slow reaction kinetics, precursor volatility, amorphization and computational inaccuracy as four broad categories of 'failure modes' that prevented the synthesis of these targets. The prevalence of each failure mode is shown in Fig. 4, accompanied by their affected targets.

Sluggish reaction kinetics hindered 11 of the 17 failed targets, each containing reaction steps with low driving forces (<50 meV per atom; Supplementary Fig. 4). In principle, these targets can be made accessible by using a higher synthesis temperature, longer heating time, improved precursor mixing or intermittent regrinding—standard procedures that are at present outside the domain of the A-Lab's active-learning algorithm. As such, we manually reground the original synthesis products generated by the A-Lab and heated them to higher temperatures, which led to the successful formation of two further targets, $Y_3Ga_3In_2O_{12}$ and $Mg_3NiO_4$, bringing our total success rate to 67% (Supplementary Note 7). One could also use more reactive precursors

to provide a greater driving force to form the target, although our experiments were constrained to air-stable binary precursors that sometimes restricted the A-Lab's choice of synthesis routes to those forming highly stable intermediates. System modifications to enable multistep heating, intermediate regrinding and expanded precursor selection should improve the ability of the lab to adapt and overcome failed synthesis attempts.

Precursor volatility disrupted all synthesis experiments targeting $CaCr_2P_2O_9$, causing a change in the net stoichiometry of its samples (Supplementary Note 8). This can be attributed to the use of ammonium phosphate precursors, $NH_4H_2PO_4$ and $(NH_4)_2HPO_4$, which proceed through a series of decomposition reactions and ultimately evaporate above 450 °C (ref. [37]). Still, recipes based on these precursors can succeed if the ammonium phosphate reacts with another precursor before its evaporation temperature, effectively locking the phosphate ions in the solid state. For example, volatility does not seem to be an issue for the Mn-containing phosphates targeted in this work, as each Mn oxides precursor reacts with the ammonium phosphates at low temperature (<500 °C) to form $Mn_2(PO_4)_3$ as an intermediate. This precursor behaviour can, in principle, be learned when sufficient pairwise reaction data have been collected, after which the A-Lab may favour the selection of precursors that trap in phosphate ions at low temperature and therefore preclude unwanted volatility.

Melting of samples at high temperature inhibited the crystallization of one target, $Mo(PO_3)_5$, whose synthesis attempts produced amorphous samples (Supplementary Fig. 5). Although the use of a molten flux can sometimes improve reaction kinetics[38], the formation of an amorphous state that is low in energy may reduce the driving force for crystallization. Indeed, using the workflow outlined in ref. [39], we identified amorphous configurations of $Mo(PO_3)_5$ with energies as low as 61 meV per atom above the crystalline ground state, a finding that is consistent with the widely reported glass-forming ability of phosphate-rich compounds[40,41].

Some failure modes result from inaccuracies in the computed stability of the target and therefore cannot be addressed by modifications to the experimental procedures. Fundamental-electronic-structure

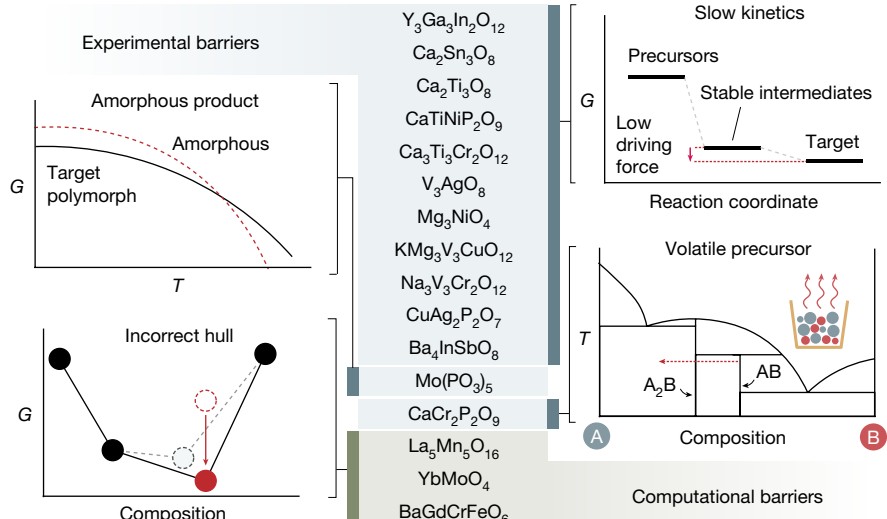

**Fig. 4 | Barriers to the synthesis of materials predicted to be stable.** The 17 target materials that could not be synthesized by the A-Lab, for which each is categorized by the feature that complicates its synthesis. One target ($Ta_4PbO_{11}$) is excluded from this list, as it is metastable and therefore was predictably unobtained in favour of its stable competitors. The challenges in synthesizing the remaining 16 stable targets fall under two categories: experimental barriers (blue, 13 targets) and computational barriers (green, three targets). We distinguish these barriers as four distinct failure modes: slow reaction kinetics, precursor volatility, product amorphization and limitations associated with DFT calculations performed at 0 K. A schematic explanation for each failure mode is provided.

challenges are probably affecting $La_5Mn_5O_{16}$, as all the attempts to synthesize this phase instead yielded $LaMnO_3$, which DFT unexpectedly predicts to be highly unstable (120 meV per atom above the hull), even though it is widely reported in the literature to be experimentally accessible[42]. If the energy of $LaMnO_3$ were lowered, consistent with its experimental stability, $La_5Mn_5O_{16}$ would be destabilized (above the hull). Errors in the computed energy of $LaMnO_3$ may arise from its strong Jahn−Teller activity[43], compositional off-stoichiometry[44] or the presence of f-states in La−all of which present challenges to conventional DFT. Problems with $YbMoO_4$ were found to be because of a poor pseudopotential choice in the Materials Project that destabilizes the well-known oxide, $Yb_2O_3$, and it is likely that, in more accurate calculations, $YbMoO_4$ is not stable. A similar lanthanide-related electronic-structure problem may also be responsible for the failure to synthesize $BaGdCrFeO_6$. These examples demonstrate the ability of the A-Lab to provide important feedback to high-throughput computed datasets. With improved calculations that exclude the computationally problematic compounds in this work, our total success rate would increase to 70% (38/54 targets).

## Outlook

In 17 days of closed-loop operation, the A-Lab performed 353 experiments and successfully realized 36 of 57 inorganic crystalline solids with diverse structures and chemistries. This high success rate (63%) was achieved by integrating robotics with: (1) DFT-computed data to survey the energetic landscape of precursors, reaction intermediates and final products; (2) heuristic suggestions for synthesis procedures obtained from ML models trained on text-mined synthesis data; (3) ML interpretation of experimental data; and (4) an active-learning algorithm that improves on failed synthesis procedures. The study also revealed several opportunities to enhance the lab's active-learning algorithm by addressing failures caused by slow reaction kinetics, which would enable an improved success rate of 67% with in-line solutions.

Our paper demonstrates that autonomous research agents can markedly accelerate the pace of materials research. Researchers initialized the A-Lab by proposing 57 target materials, which were successfully realized at a rate of >2 additional materials per day with minimal human intervention. Such rapid and successful experimentation point to a vast landscape of opportunities in materials synthesis and development. Although this work focused on a limited subset of all possible synthesis targets, many new candidates await evaluation. As the breadth of ab initio computations continues to grow, so will this list of materials.

While the characterization of multi-phase samples remains challenging, decision making in the A-Lab is supported by its robust identification of the majority phase in each sample, combined with the use of computed phase stability data. Nonetheless, there remains ample room for improvement in the automated interpretation of materials characterization data. Algorithms that can more effectively account for impurity phases in XRD patterns, augmented with additional characterization techniques such as electron microscopy and mass spectrometry, would benefit the next generation of autonomous research platforms.

Advances in simulations, ML and robotics have intersected to enable 'expert systems' that show autonomy as an emergent quality by the sum of its automated components. The A-Lab demonstrates this by combining modern theory-driven and data-driven ML techniques with a modular workflow that can synthesize materials with minimal human input. Lessons learned from continuing experiments can inform both the system itself and the greater community through systematic data generation and collection. The systematic nature of the A-Lab provides a unique opportunity to answer fundamental questions about the factors that govern the synthesizability of materials, serving as an experimental oracle to validate predictions made on the basis of data-rich resources such as the Materials Project. In future iterations of the platform, such an oracle may be expanded to investigate factors beyond synthesizability, including microstructure and device performance. Although our current success rate is high, the remaining discrepancies between current predictions and their experimental outcomes is a crucial signal required to improve our understanding of materials synthesis.

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

## Methods

### Materials screening

The 57 targets evaluated by the A-Lab were identified from the Materials Project database (version 2022.10.28). We first obtained all entries from the Materials Project that were marked as 'theoretical' (that is, not represented in the ICSD) and predicted to be thermodynamically stable (at 0 K) or very close to the convex hull (<10 meV per atom). We did not consider materials with ≤2 elements nor those containing elements that are radioactive (Ac, Th, Pa, U, Np, Pu, Tc), exceedingly rare (Pd, Pt, Rh, Ir, Au, Ru, Os, Re, Tl, Sc, Tm, Pm, Rb, Cs) or toxic (Hg, As). Owing to concerns with the experimental handling of certain materials systems (for example, sulfides), we constrained our selection to only include the following types of material: oxides, carbonates, bicarbonates, hydroxides, sulfates, sulfites, bisulfates, silicates, fluorides, chlorides, bromides, orthoborates, metaborates, tetraborates, phosphates, phosphites, chlorates, chlorites and hypochlorites. Finally, we removed all compounds predicted to have uncommon and potentially challenging oxidation states (for example, $Co^{4+}$), as determined by pymatgen[45].

We ensured that each candidate material was new to the A-Lab by cross-checking with several experimental sources. We first removed all compositions that appeared in SynTERRA, a text-mined set of experimental synthesis data extracted from more than 24,000 publications[46]. Furthermore, we removed any materials with compositions appearing in the 'Handbook of Inorganic Substances'[47]. For the remaining 432 candidates identified using this workflow, we filtered by thermodynamic stability in air. This was done by calculating the formation energy of each compound in a grand potential with respect to oxygen, assuming standard atmospheric conditions ($p_{O_2} = 21{,}200$ Pa) and temperatures ranging from 600 to 1,100 °C. We further checked for reactivity with $CO_2$ and $H_2O$ under those same conditions by using the Interface Reactions module in pymatgen[45,48]. From the resulting list of 146 compounds that were stable in air, we selected 57 targets for which precursors were readily available. Notably, there are previous experimental reports of the same or closely related compositions for many targets (Supplementary Table 3). However, these reports were not included in the algorithms that the A-Lab uses to design its initial guess for a synthesis recipe.

The algorithm we used for identifying potential synthesis targets is available on GitHub (https://github.com/mattmcdermott/novel-materials-screening). It operates autonomously once given the following information: which elements to consider in the target materials, how large an upper limit to impose on each material's energy above the convex hull, the atmospheric conditions under which the materials will be synthesized and a threshold on the reaction energies that exist between each material and the gaseous species present in the specified atmosphere. The algorithm then scrapes the Materials Project and produces a list of candidate materials that satisfy these criteria. Further filtering may be considered on the basis of the availability and cost of precursors for each target. Although this is done manually in the current version of the algorithm, potential improvements could automate the process by using online data from chemical inventory lists and vendor websites.

### Synthesis recipes from text-mined knowledge

We have established a pipeline for recommending synthesis recipes by using a knowledge base of 33,343 solid-state synthesis procedures extracted from 24,304 publications[23]. For a given target, the initial recipe is selected on the basis of the most common precursors in the knowledge base. We then transition to a similarity-based strategy for recipe selection. Each target is transformed into a numerical vector by using a synthesis-context-based encoding model[22]. The similarity between a given target and each material in the knowledge base is evaluated using the cosine similarity between their encoded vectors. After identifying the reference material that is most similar to the target, its precursors are included in the new recommendation. When these precursors do not cover all the elements in the target, we use a masked precursor completion model[22] to account for such missing precursors. Subsequent recommendations are implemented by moving down the list of known materials ranked to be most similar to the target.

For each set of recommended precursors, the most effective synthesis temperature is predicted using an XGBoost regressor trained in previous work[23]. The target and its associated precursors are transformed into three sets of features: (1) precursor properties including melting points, standard enthalpies of formation and standard Gibbs free energies of formation; (2) target compositional features indicating which elements are present; and (3) the calculated thermodynamic driving force associated with pairwise reaction paths from precursors to target. Although the proposed synthesis temperature is dependent on the precursors, not just the target, it may vary for each recipe. However, to maximize the efficiency with which the A-Lab operates, we chose to use one fixed temperature for each target. This temperature was calculated by averaging the proposed synthesis temperatures for the top five precursor sets recommended for a given target. This allowed all such precursor sets to be batched in a single furnace.

### Robotic synthesis and characterization

The A-Lab performs fully automated solid-state synthesis and characterization. It is a bespoke robotic platform that consists of a precursor preparation station with a central robot arm (Mitsubishi) for powder dispensing and mixing (custom-made with Labman Automation Ltd.), a high-temperature heating station with four box furnaces (based on F48055-60, Thermo Scientific, with custom actuators to control its door), a product-handling station developed in-house for powder retrieval and sample loading, a characterization station with a powder X-ray diffractometer (Aeris Minerals, Malvern Panalytical) and two collaborative robot arms (UR5e, Universal Robots) that transfer samples and labware between stations. Further details on the robotic platform are provided in Supplementary Note 9.

The synthesis process starts from the precursor preparation station, where the necessary consumables (plastic vials, $ZrO_2$ balls and crucibles) and precursor dosing bottles containing between 50 and 100 g of powders are manually loaded before starting a new experimental campaign. Prescribed amounts of the precursor powders are dispensed into a plastic vial by an automatic dispenser balance (Quantos, Mettler Toledo). The precursor powders are then mixed thoroughly with ethanol and ten 5-mm $ZrO_2$ balls in a dual asymmetric centrifuge (Smart DAC250, Hauschild) for 9 min. To ensure proper slurry viscosity, the ethanol amount is automatically calculated on the basis of the amount and density of each powder comprising the mixture. The resulting slurry is transferred with an automated pipettor (rLine LH-710969, Sartorius) into an alumina crucible, which is then dried at 80 °C in a closed evaporation system. A UR5e robot arm on a linear rail (Olympus Controls) removes the dried samples from the precursor preparation station and loads them into one of four box furnaces. Heating is performed in batches, with each furnace containing up to eight samples on an alumina tray. Each batch is heated to 300 °C with a slow ramping rate of 2 °C min$^{-1}$ to raise the likelihood that any phosphate precursor has time to react before it becomes volatile at higher temperature. The samples are then heated to the specified synthesis temperature with a nominal ramp rate of 15 °C min$^{-1}$, followed by a 4-h dwell. After the dwell is complete, the samples are naturally cooled to 100 °C, at which point a UR5e arm removes the samples from the furnace and waits another 10 min to allow the samples to cool to room temperature.

A separate UR5e arm transfers the cooled samples to the next station for powder retrieval and characterization. There, a 10-mm alumina ball is placed in each crucible by an automatic ball dispenser developed in-house and then sent to a vertical shaker that grinds the samples into fine powders. The resulting powders are then poured by the UR5e arm from the crucibles into a clean plastic vial covered using a steel mesh. By inverting the container, the powder is dispensed through

the mesh onto an XRD sample holder and subsequently flattened with an acrylic disc. The UR5e arm transfers each flattened sample into the diffractometer for X-ray measurements, which are performed using 8-min scans that range from $2\theta = 10°$ to $100°$. The XRD sample holders must be cleaned manually when the lab has depleted its stock. Precursor powders should also be refilled or replaced, when necessary, although this can be performed without stopping the workflow of the lab. Similarly, hardware exceptions can be handled manually in the affected station without interrupting operations in the rest of the lab. The average exception rate across all stations in the A-Lab is about 3.9% over 1.5 years of operation, with details provided in Supplementary Note 10 and Supplementary Fig. 6.

## Phase analysis

Given an XRD pattern obtained from an unknown sample, we apply XRD-AutoAnalyzer to identify the constituent phases and estimate their weight fractions[25]. This algorithm relies on a convolutional neural network (CNN) consisting of six convolutional layers, with max pooling applied between each, followed by three fully connected layers with ReLU activation. Batch normalization and a dropout rate of 50% is applied between the fully connected layers for regularization. At inference, we apply Monte Carlo dropout to create an ensemble of 100 networks with 50% of their connections randomly excluded. The final prediction is taken as the phase that seems most frequently in the ensemble and its associated confidence is defined as the fraction of models that predict it.

A unique model instance is trained on the chemical space defined by each target. Experimental-structure entries with elements shared by the given target are extracted from the ICSD, also including carbonates and hydroxides. For the DFT-calculated target, we apply a machine-learned volume correction to its lattice parameters (Supplementary Note 1) before including it in the training set. From each reference phase, 200 diffraction patterns are simulated with stochastic variations derived from experimental artefacts including lattice strain, crystallographic texture, impurity peaks and poor crystallinity. These augmented patterns are used to train the CNN for 50 epochs, after which they are ready for the analysis of novel patterns.

To confirm the predictions of the CNN, we use an automated approach to multiphase Rietveld refinement. An agent with two deep neural networks (actor/critic) were trained using reinforcement learning based on a proximal policy optimization algorithm[49] implemented in a custom gym environment[50] that interacts with the GSAS-II software package[51] through a scripting interface[52] (Supplementary Note 2). The environment is initialized by refining the background, followed by the scale factor and sample displacement. After initialization is performed on the basis of these parameters, the algorithm freely refines the lattice parameters, phase fractions, isotropic microstrains and particle sizes. For each step in the refinement, our algorithm decides which parameters to refine and/or reset to the initial values, with the objective of minimizing the difference between the calculated and the experimentally observed patterns.

When the automated refinement gives a poor fit, manual analysis is performed. For targets for which we suspect the poor fit resulted from configurational disorder, we refined the XRD patterns using cation-disordered versions of the target's structure taken from the Materials Project. The cations allowed to be exchanged (disordered) with one another were selected on the basis of the Hume-Rothery rules, as detailed in previous work[25]. Such cases were still considered successful as long as the disordered version of the target retained the same crystal structure and overall composition as the ordered version.

To confirm that the A-Lab made 36 of the 57 targets considered in this work, we manually performed Rietveld refinement on the XRD data that was acquired from the original (autonomous) experiments. This reanalysis served only as external validation of the claimed synthesis outcomes, and therefore did not contribute to A-Lab's decision making.

In the manual refinement, we checked whether the diffraction peaks associated with the XRD pattern could be fit by any known phase other than the target. When such phases existed, the quality of their fit to the experimental pattern was compared with the quality of the target's fit. These refinements may include additional secondary phases that were manually identified by referring to experimental databases such as the ICSD and ICDD. The best fits achieved through this process are provided in the Supplementary Data. They are accompanied by lattice parameter analysis to estimate the compositions of any targets that are suspected to form solid solutions.

## Active-learning algorithm

Active learning is performed using ARROWS[3], our recently developed algorithm that learns from previous experimental outcomes to identify improved reaction pathways. Given the products obtained from a set of precursors proposed by our natural-language models at temperature $T_{NLP}$, ARROWS[3] first suggests that a lower temperature ($T_{NLP} - 300$ °C) be tested for the same precursor set. The intent of this approach is to reveal which intermediate phases lead to the formation of each impurity observed at higher temperature. From the low-temperature-synthesis outcome, information is extracted about the pairwise reactions that occurred, including those between the precursors (to form the observed intermediates), as well as those between the intermediates (to form the high-temperature impurities). New synthesis experiments are then proposed on the basis of sets of precursors expected to avoid such reactions, giving priority to those with a maximal thermodynamic driving force to form the target. The driving force is calculated as the free-energy difference between a target and its associated precursors, in which all solid energies (at 0 K) are extracted from the Materials Project and corrected using a machine-learned descriptor that accounts for vibrational-entropy contributions at the specified temperature[53].

After testing a precursor set at low temperature ($T_{NLP} - 300$ °C), iteratively higher temperatures ($\Delta T = 100$ °C) are examined until the target is obtained with a yield exceeding 50% or until the temperature reaches $T_{NLP}$. At each step, the algorithm determines which pairwise reactions occurred and records them in a database that is referred to throughout all other experiments performed by the A-Lab. In subsequent iterations, ARROWS[3] gives priority to sets of precursors containing pairs of phases that are expected to form the desired target, while avoiding pairs that form unwanted impurities. Moreover, to avoid testing redundant synthesis routes for which different precursors form identical products, the algorithm checks whether the low-temperature ($T_{NLP} - 300$ °C) intermediates obtained from a given precursor set differ from those obtained with previous (unsuccessful) recipes. If not, then no further experiments are proposed for that set of precursors. This process is repeated until the target is successfully obtained or until all the available precursor sets have been exhausted. Further details on the active-learning process are provided in Supplementary Notes 4–6 and 11.

## Data availability

All data generated during this study are included in the Supplementary Information. This includes the refined XRD patterns for each successful synthesis outcome, as well as their associated structure files. This includes the XRD patterns acquired from reported successes that were refined with and without manual intervention, in addition to their associated structure files.

## Code availability

The screening algorithm we used for identifying potential synthesis targets is available at https://github.com/mattmcdermott/novel-materials-screening. The Python scripts and machine-learning

models used to propose literature-inspired synthesis recipes can be found online at https://github.com/CederGroupHub/SynthesisSimilarity and https://github.com/CederGroupHub/s4 for precursor and temperature selection, respectively. The methods for XRD analysis are available at https://github.com/njszym/XRD-AutoAnalyzer. Active learning was performed using a package found at https://github.com/njszym/ARROWS.

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

**Acknowledgements** This work was primarily financed by the U.S. Department of Energy, Office of Science, Office of Basic Energy Sciences, Materials Sciences and Engineering Division under contract no. DE-AC02-05-CH11231 (D2S2 programme, KCD2S2) and the Laboratory Directed Research and Development Program of Lawrence Berkeley National Laboratory. Development of the active-learning algorithms, compound-discovery methods and equipment acquisition were supported by the Materials Project programme (KC23MP). Machine-learning techniques for the interpretation of XRD patterns were developed by the Joint Center for Energy Storage Research programme JCESR 2.0 under contract no. DE-AC02-05-CH11231. Computations were performed using the National Energy Research Scientific Computing Center (NERSC), a U.S. Department of Energy Office of Science User Facility supported by the Office of Science and the U.S. Department of Energy under contract no. DE-AC02-05CH11231. Work done at UC Berkeley was supported by Umicore Specialty Oxides and Chemicals. N.J.S. was supported in part by the National Science Foundation Graduate Research Fellowship under grant no. 1752814. We thank Labman Automation for their role in the design and construction of hardware for precursor preparation. We also thank M. Sargent at Berkeley Lab for capturing photos of the A-Lab.

**Author contributions** N.J.S. developed the algorithms for data analysis and decision-making. N.J.S. and B.R. developed the methods used to correct DFT-calculated lattice parameters. B.R. and Y.F. built the lab's hardware and developed the refinement algorithm. R.E.K. and Y.F. designed the control software and its integration with the hardware. T.H. created the algorithms for literature-inspired synthesis recipe recommendation. D.M. assisted in hardware development. M.J.D. and M.G. built the filtering pipeline for candidate screening. E.D.C. and A.M. applied the filtering from Google DeepMind. C.J.B. assisted in planning the A-Lab's setup, developing the algorithms for analysis and decision-making, and modelling the lab's throughput. H.K. and A.J. supervised hardware and software development, respectively. K.P. supervised the contributions from the Materials Project. Y.Z. and G.C. conceived and supervised all of the main aspects of the project.

**Competing interests** The authors declare no competing interests.

**Additional information**
**Correspondence and requests for materials** should be addressed to Yan Zeng or Gerbrand Ceder.

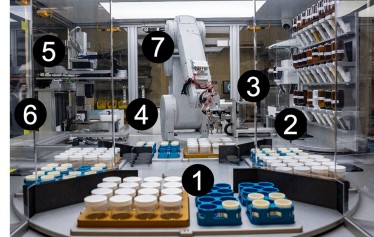 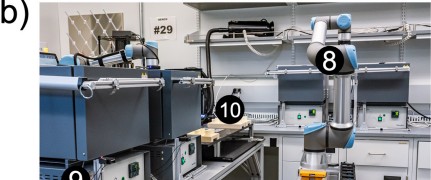 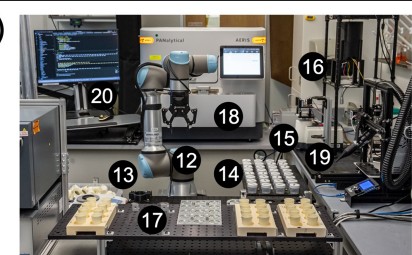

**a) Precursor preparation**
1. Loading/unloading
2. Powder dispensing
3. Solvent addition
4. Mixing
5. Slurry transferring
6. Slurry drying
7. Robot arm R1

**b) Heating**
8. Robot arm R2
9. Box furnaces
10. Crucible racks
11. Handle

**c) Product handling and characterization**
12. Robot arm R3
13. Weighing balance, capper
14. Sample storage
15. Vertical shaker
16. Ball dispenser
17. XRD sample holders
18. X-ray diffractometer
19. Vacuum cleaner
20. Central control PC

**Extended Data Fig. 1 | A-Lab hardware setup.** Detailed overview of all physical components in the A-Lab, including the stations for precursor preparation, heating and product handling for XRD characterization.

a)

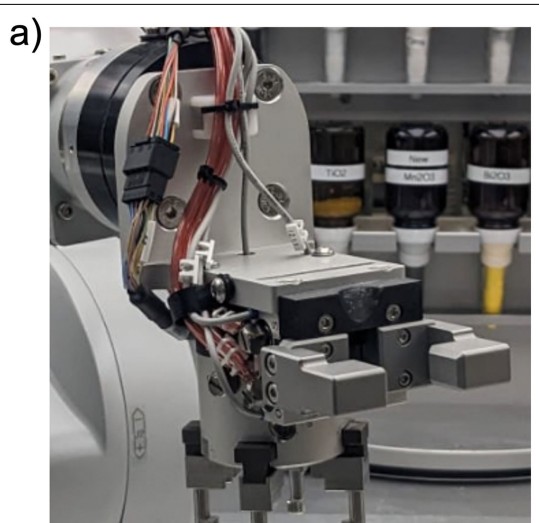

b)

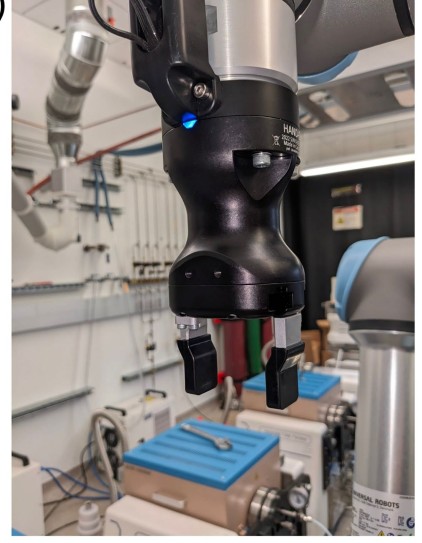

c)

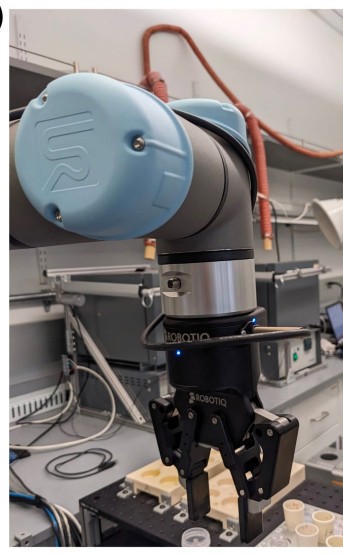

d)

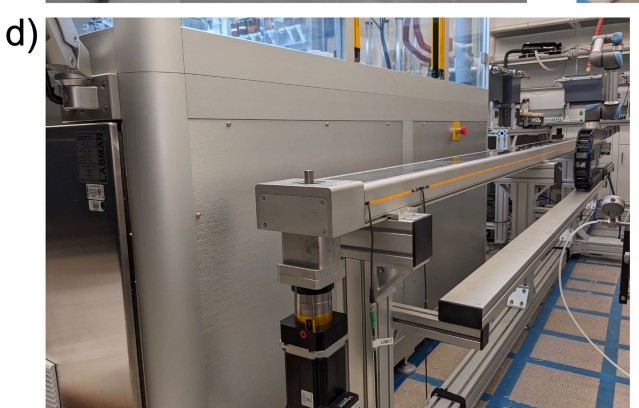

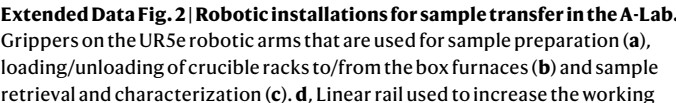

e)

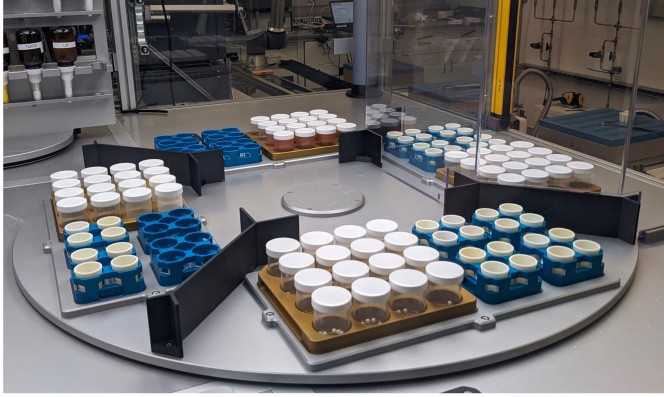

**Extended Data Fig. 2 | Robotic installations for sample transfer in the A-Lab.**
Grippers on the UR5e robotic arms that are used for sample preparation (**a**),
loading/unloading of crucible racks to/from the box furnaces (**b**) and sample
retrieval and characterization (**c**). **d**, Linear rail used to increase the working
envelope of the robotic arm that loads/unloads crucible racks to/from the
furnaces. **e**, Carousel used to organize and move samples in the sample
preparation station.

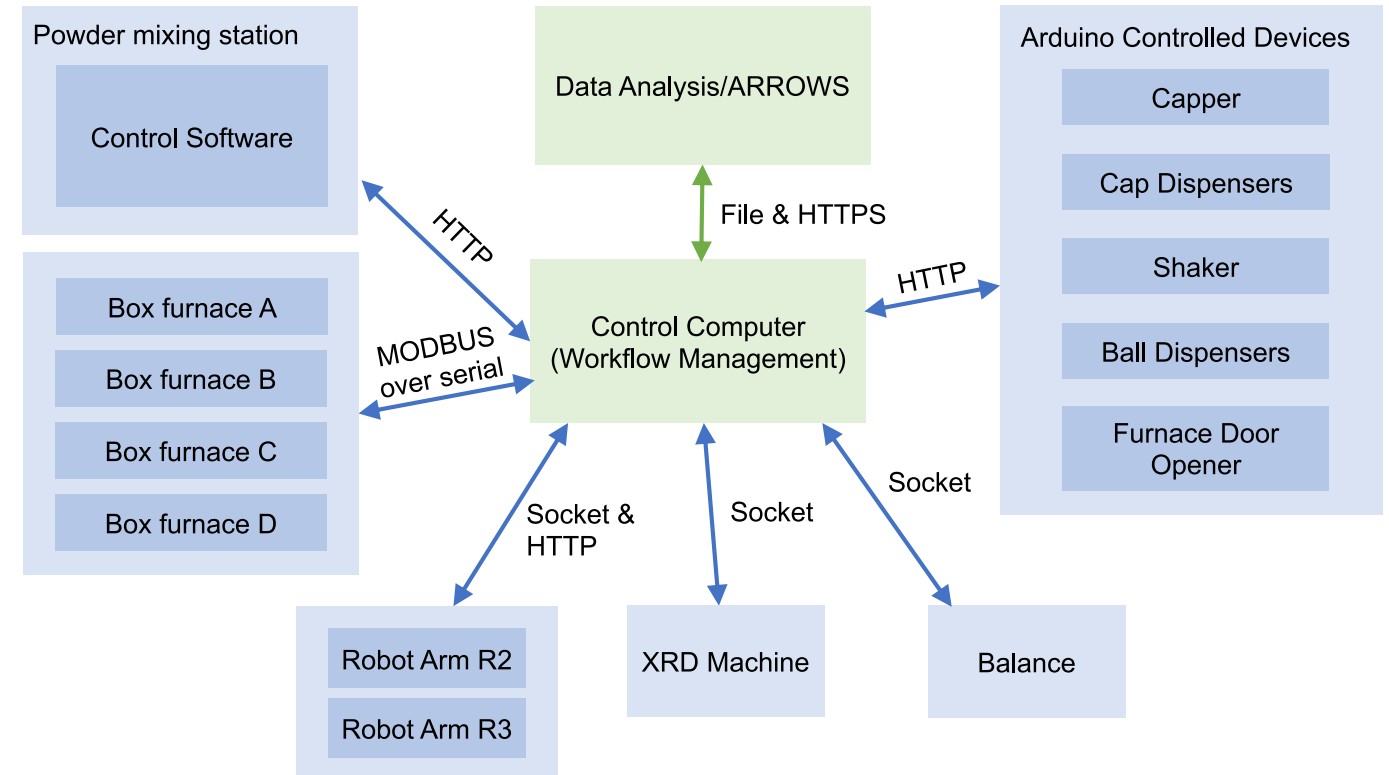

**Extended Data Fig. 3 | Communication protocols connecting each module in the A-Lab.** A local area network (LAN) is built to connect all the pieces of the A-Lab with a control computer using an RS-485 interface (or DB25 for the box furnaces). Each module on the RS-485 interface has an Internet Protocol (IP) assigned to enable communication with the computer. For enhanced cybersecurity, only the control computer has access to the internet, whereas the LAN is isolated from it.