## [Peer Review File · Nature]

Manuscript Title: An autonomous laboratory for the accelerated synthesis of novel materials

Reviewer Comments & Author Rebuttals

Reviewer Reports on the Initial Version:

Referees' comments:

Referee #1 (Remarks to the Author):

In this paper, the authors proposed an autonomous lab that integrates robotics with the use of ab-initio databases, machine learning-driven data interpretation, synthesis heuristics learned from text-mined literature data, and active learning to develop synthesis routes for novel inorganic materials in the solid state. The authors first proposed 58 target materials by searching the ab-initio database and some other constraints in synthesis which were not experimentally synthesized previously but predicted to be thermodynamically stable from computations. Then the authors showed that within 17 days the A-Lab successfully developed synthesis routes for 41 novel compounds from the 58 target materials, with synthesis recipes proposed by natural language models trained on the literature and optimized using an active learning approach grounded in thermodynamics. The authors also analyzed failed syntheses, which enables them to provide direct and actionable suggestions to improve current techniques for materials screening and synthesis design.

As the materials synthesis is traditionally based on trial-and-error style, the efficiency of successful novel materials discovery is typically low. In the past one and two decades, with the improved accuracy of density functional theory calculations and more powerful computers, high-throughput computational screening and the associated ab-initio database can provide candidate materials of some target properties for experimental synthesis, and thus accelerate discoveries of novel materials in comparison with the traditional approach. However, the gap between the number of theoretical candidates and that of experimental realization is still typically large. This paper makes one step further by connecting the theoretical candidates with experimental realizations with synthesis recipes from text mining of existing literature and experimental characterizations that can be improved using active learning and unified within a robotic platform. The A-Lab not only delivers excellent successful synthesis rate, but also is able to analyze the failed attempts and improve synthesis routes for some target materials to be realized experimentally, both of which are very impressive. Most importantly, the A-Lab is autonomous. This suggests that the A-Lab is promising for materials discoveries and further accelerates synthesis of novel inorganic materials. Also, the paper is well written and easy to follow. I therefore recommend for publication.

There are two minor suggestions/comments:

1. For each supplementary note, it might be helpful to readers if a summarizing title is given.
2. For the failure to synthesize BaGdCrFeO₆, could it be due to some entropic effects that can not be included in DFT calculations since it has 5 elements and can be regarded as a high-entropy compound.

Referee #2 (Remarks to the Author):

This is an exciting, very nicely written research article which showcases the possibilities of autonomous experiments for the discovery of new materials. By screening a large database of

predicted compounds feasible candidates were selected and new compounds were synthesized with a high success rate in an autonomously working lab setup. This work is novel and highly significant for the accelerated development of new materials.

I suggest to address the following issues:

In my opinion, the title is too general by using the term "inorganic materials". The impressive results of the article are shown exclusively for oxides and also one specific synthesis and processing route leading to powders. What about metals, alloys and intermetallics? What about other materials states like bulk materials or thin films. Are the results directly applicable to these material classes? If yes, the authors should add evidence and explanations for this. Or if this is currently not possible, change the title. Instead of materials: oxide powders. Specific experiments on powders (precursors, heating in air) were performed and the authors mention "the unique challenges of handling and characterizing solid inorganic powders". It should be explained what is "unique" to this.

The authors mention several times the total duration of the experiments which was "17 days of uninterrupted operation". Here I have some questions: How much preparation days were necessary to perform this successful 17-days, or was this the first try? How many failed runs before this?

How sensitive is the success rate on the pre-selection of targets? The pre-selection of the 58 targets was done by researchers, not autonomously. If other targets would have been chosen, maybe the success rate would have been lower or higher? Could the pre-selection also be performed autonomously in the future?

Can the authors give a prediction what their results means for the future? If the impressive success rates would hold true for other materials classes: Are we done with discoveries in a few years? Or what do you rate as limits?

From the materials screening section it reads a bit like that there is not much more to discover as you screened all theoretical predictions in the Materials Project and only 146 feasible candidates were left from which you showed in 17 days what can be synthesized. I don't think that is the message which should be conveyed. The authors could consider adding some sentences where they explain the possibilities which are still open in the materials search space and put their approach in the right perspective.

Any perspective on microstructure, which is also highly relevant for useful materials properties? In other words: it is not sufficient to "only" discover new materials but their microstructure is also highly important.

The concept of decomposition energies should be explained a bit more, earlier in the text. Also, the meaning of targets in this article should be directly explained in the main text when it first appears.

Where words such as paradigm, synergy, unique are used, it would be good to explain what they exactly mean in the context of the article.

The authors could consider to explain more in the main text on the data interpretation and decision making abilities of the experimental agent.

For multi-component compounds: Is pairwise reaction analysis (Fig. 3) sufficient?

Fig. 4: Not clear what frequencies means here, cannot find it in the Fig.

Referee #3 (Remarks to the Author):

This paper presents an autonomous laboratory for the accelerated synthesis of novel inorganic materials, and demonstrate this with the synthesis of 41 novel compounds (out out of a set of 58 targets).

The work is ambitious in its concept, aiming to go beyond previous autonomous materials experiments (e.g., refs 3-6 in the MS) by folding in literature mining and computational predictions. To some degree, most of the individual components in the overall method are known but bringing all of them together in a successful experiment is non-trivial indeed, and outcome of the work (41 new compounds) is really impressive. I feel it is the kind of proof-of-concept of a new way of doing research that might fit in a journal such as this.

My main criticisms would be presentational - I found the manuscript a little difficult to review and quite hard to follow. This is, in part, because the authors are trying to pack a lot of concepts (automation, chemistry, algorithms, literature scraping) into a rather short format, but nonetheless, I genuinely doubt that researchers who are not close to this area will find the paper easy to digest, at least beyond the high-level concepts, aims, and outcomes (which are clear enough). This is exacerbated, I think, by the fact that the Supporting Information is not very long or detailed, as normalised to the complexity of what has been done. I have seen papers in this general area with far more technical detail in the ESI, and I think for such new methods this is essential. At times, it almost felt like the paper read as a summary or highlight, rather than a research paper. I don't think this is really the fault of the authors, who are excellent communicators, it is really just a symptom of attempting to explain a really complex and multifaceted study in rather little space. Length constraints aside, I think there are some ways that the paper might be restructured to address these issues. For example:

* Give at least one 'walked-through example' of how a single material was discovered. There are some details about specific compounds in the main text (e.g., $\text{CaFe}_2\text{P}_2\text{O}_9$, $\text{Mo}(\text{PO}_3)_5$, $\text{La}_5\text{Mn}_5\text{O}_{16}$), but they are short, and it rather ends up reading like a potpourri of asides. This is partly because the authors report so many new compounds, which is impressive, but it makes for a hard read if you really want to understand the methodology. This could be backed up by other detailed 'walk throughs' in the ESI - I wouldn't expect that for all compounds, but it would be really useful to flesh out the approach, and could include some 'failed' recipes, too, to highlight where things can go wrong.

*Provide more detail about the active learning parts. The experimental aspects are clear enough and figure 1 shows the workflow (powder dose  heat  XRD), but the recipe generation / choice aspects, notwithstanding Figure 2, were harder to follow.

* While the overall experimental scheme is clear enough (Figure 1), some of the details are not. For example, the description of the automated synthesis and PXRD process, which is complicated, could be made clearer and augmented with explanatory figures in the ESI.

Author Rebuttals to Initial Comments:

Reviewer #1

General Assessment:

In this paper, the authors proposed an autonomous lab that integrates robotics with the use of ab-initio databases, machine learning-driven data interpretation, synthesis heuristics learned from text-mined literature data, and active learning to develop synthesis routes for novel inorganic materials in the solid state. The authors first proposed 58 target materials by searching the ab-initio database and some other constraints in synthesis which were not experimentally synthesized previously but predicted to be thermodynamically stable from computations. Then the authors showed that within 17 days the A-Lab successfully developed synthesis routes for 41 novel compounds from the 58 target materials, with synthesis recipes proposed by natural language models trained on the literature and optimized using an active learning approach grounded in thermodynamics. The authors also analyzed failed syntheses, which enables them to provide direct and actionable suggestions to improve current techniques for materials screening and synthesis design.

As the materials synthesis is traditionally based on trial-and-error style, the efficiency of successful novel materials discovery is typically low. In the past one and two decades, with the improved accuracy of density functional theory calculations and more powerful computers, high-throughput computational screening and the associate ab-initio database can provide candidate materials of some target properties for experimental synthesis, and thus accelerate discoveries of novel materials in comparison with the traditional approach. However, the gap between the number of theoretical candidates and that of experimental realization is still typically large. This paper makes one step further by connecting the theoretical candidates with experimental realizations with synthesis recipes from text mining of existing literature and experimental characterizations that can be improved using active learning and unified within a robotic platform. The A-Lab not only delivers excellent successful synthesis rate, but also is able to analyze the failed attempts and improve synthesis routes for some target materials to be realized experimentally, both of which are very impressive. Most importantly, the A-Lab is autonomous. This suggests that the A-Lab is promising for materials discoveries and further accelerates synthesis of novel inorganic materials. Also, the paper is well written and easy to follow. I therefore recommend for publication.

Response:

We thank the reviewer for their positive comments. Each suggested change is addressed below.

Comment 1:

For each supplementary note, it might be helpful to readers if a summarizing title is given.

Response:

A title has been added to each Supplementary Note. These titles are summarized below.

Supplementary Notes:

1. Correction of DFT-calculated lattice parameters
2. Automated Rietveld refinement
3. Literature-inspired synthesis recipe generation for $\text{MgNi}(\text{PO}_3)_4$
4. Identification of unique synthesis pathways
5. Exhausting the search space for BaGdCrFeO_6 synthesis
6. Successful optimization of $\text{CaFe}_2\text{P}_2\text{O}_9$ synthesis
7. Synthesis modifications to overcome slow reaction kinetics
8. Characterization of precursor volatility
9. Assessing the novelty of target materials
10. Specifications of robots in the A-Lab
11. Exceptions to pairwise reaction analysis

Comment 2:

For the failure to synthesize BaGdCrFeO_6 , could it be due to some entropic effects that cannot be included in DFT calculations since it has 5 elements and can be regarded as a high-entropy compound?

Response:

While configurational entropy can indeed affect the free energy of disordered materials, BaGdCrFeO_6 is predicted to adopt a completely ordered structure wherein each cation occupies a distinct crystallographic site. Even if it were to disorder, the configurational entropy would only serve to further stabilize the compound, and therefore it cannot be the reason as to why BaGdCrFeO_6 failed to be synthesized.

We would also like to point out that the decomposition energy of BaGdCrFeO_6 is computed to be very large (-2443 meV/atom) and therefore its predicted stability is unlikely to be affected by configurational entropy, whose contributions to the free energy are typically on the order of 10-100 meV/atom.

Reviewer #2

General Assessment:

This is an exciting, very nicely written research article which showcases the possibilities of autonomous experiments for the discovery of new materials. By screening a large database of predicted compounds feasible candidates were selected and new compounds were synthesized with a high success rate in an autonomously working lab setup. This work is novel and highly significant for the accelerated development of new materials.

Response:

We thank the reviewer for their feedback. All questions and suggested revisions are addressed below.

Comment 1:

In my opinion, the title is too general by using the term “inorganic materials”. The impressive results of the article are shown exclusively for oxides and also one specific synthesis and processing route leading to powders. What about metals, alloys and intermetallics? What about other materials states like bulk materials or thin films. Are the results directly applicable to these material classes? If yes, the authors should add evidence and explanations for this. Or if this is currently not possible, change the title. Instead of materials: oxide powders.

Response:

In accordance with the character limitations set by the journal, we have removed “inorganic” from the title of our manuscript. We have also made changes throughout the abstract and introduction to more clearly state which types of materials the platform can be applied to: air-stable inorganic powders.

We believe that the robotic methods and AI/ML algorithms developed in our work could be transferred to materials that are not air-stable, though this would require handling of the synthesized materials (and likely the precursors) in glove boxes. Metals could in principle be handled by starting from powders.

The changes made to the manuscript are copied below.

Abstract, Page 1: To close the gap between the rates of computational screening and experimental realization of novel materials^{1,2}, we introduce the A-Lab, an autonomous laboratory for the solid-state synthesis of inorganic powders. This platform leverages computations, historical data from the literature, machine learning, and active learning to plan and interpret the outcomes of experiments performed using robotics. Over 17 days of continuous operation, the A-Lab realized 41 novel compounds from a set of 58 targets including a variety of oxides and phosphates that were identified using large-scale ab-initio phase stability data from the Materials Project and Google Brain.

Introduction, Page 2: In this work, we present the A-Lab, an autonomous laboratory that integrates robotics with the use of ab-initio databases, ML-driven data interpretation, synthesis heuristics learned from text-mined literature data, and active learning to optimize the synthesis of novel inorganic materials in powder form.

Introduction, Page 2: Given a set of air-stable target materials (*i.e.*, desired synthesis products whose yield we aim to maximize) screened using the Materials Project¹⁴, the A-Lab generates synthesis recipes using ML models trained on historical data from the literature, then executes these recipes with robotics.

Comment 2:

Specific experiments on powders (precursors, heating in air) were performed and the authors mention “the unique challenges of handling and characterizing solid inorganic powders”. It should be explained what is “unique” to this.

Response:

Most efforts to automate the synthesis of inorganic compounds have started from water-soluble precursors, which are easy to mix but tend to produce small quantities of materials while also limiting the chemistries that can be used. Solid-state synthesis is generally more versatile and can produce multi-gram quantities; however, handling a wide range of solid powders is made difficult by the fact that each compound exhibits a different set of physical properties related to its density, flow behavior, particle size, hardness, and compressibility. Solid powders also tend to require milling to ensure intimate mixing and good reactivity between the precursors (as is done in the A-Lab). These requirements warrant the use of hardware that can adapt to the specific properties of the powder at hand.

The unique challenges of handling solid powders are now specified in the main text (copied below).

Introduction, Page 2: While autonomous workflows based on liquid handling have been demonstrated in organic chemistry¹⁰⁻¹³, the A-Lab addresses the unique challenges of handling and characterizing solid inorganic powders. These often require milling to ensure good reactivity between precursors, which can have a wide range of physical properties related to differences in their density, flow behavior, particle size, hardness, and compressibility.

Comment 3:

The authors mention several times the total duration of the experiments which was “17 days of uninterrupted operation.” Here I have some questions. How much preparation days were necessary to perform this successful 17-days, or was this the first try? How many failed runs before this?

Response:

For the current project, we spent several weeks planning which targets and precursors to consider. This involved the development of the screening workflow detailed in the main text, the selection of appropriate target criteria (*e.g.*, air stability and limits on the energy above hull), and the application of machine learning models to generate precursor lists. After ordering and receiving all the precursor powders, we manually filled dosing heads that were loaded into the A-Lab along with the necessary consumables including plastic containers with ZrO₂ balls for mixing and alumina crucibles for heating. From this point on, the system required minimal human intervention aside from the refilling of precursor dosing heads and the cleaning of reusable consumables.

Prior to the execution of this project, nearly three years were spent developing the hardware and its associated algorithms. About two additional months were spent testing the robustness and repeatability of experiments performed in the lab involving well-studied chemistries. However, we stress that the experiments reported in our manuscript correspond to the first time in which the A-Lab attempted the 58 target materials being considered. There were no prior runs on any of these targets.

Both points are now clarified in the main text.

Robotic synthesis & characterization, Page 17: The synthesis process starts from the precursor preparation station, where the necessary consumables (plastic vials, ZrO₂ balls, and crucibles) and precursor dosing bottles containing between 50 and 100 g of powders are manually loaded prior to starting a new experimental campaign.

Robotic synthesis & characterization, Page 18: The UR5e arm transfers each flattened sample into the diffractometer for X-ray measurements, which are performed using 8-min scans that range from $2\theta = 10^\circ$ to 100° . The XRD sample holders must be cleaned manually when the lab has depleted its stock. Precursor powders should also be refilled or replaced, when necessary, though this can be performed without halting the lab's workflow.

Autonomous materials discovery platform, Page 4: All target materials considered in this work are new to the lab, *i.e.*, not present in the training data for the algorithms it uses to propose synthesis recipes, and 52 of the 58 targets have no prior synthesis reports to the best of our knowledge (Methods). The experiments reported in this study represent the first attempts by the A-Lab to synthesize any of these targets.

Comment 4:

How sensitive is the success rate on the pre-selection of targets? The pre-selection of the 58 targets was done by researchers, not autonomously. If other targets would have been chosen, maybe the success rate would have been lower or higher? Could the pre-selection also be performed autonomously in the future?

Response:

As detailed in the **Methods**, we applied strict (and automated) filtering to identify 146 targets for the current project. This automated screening process included criteria such as the hull energy, computed reactivity with O₂, H₂O, and CO₂ (so that samples could be handled in air), and the exclusion of toxic or expensive elements. From this list, we manually selected a subset of 58 compounds for which precursors were already available in our laboratory or which could be readily purchased from vendors. This could in principle be automated in future iterations of the screening algorithm, *e.g.*, by coupling it with existing chemical management systems that track inventory in the lab. Furthermore, one could envision an algorithm that scrapes data from the websites of common vendors such as Sigma Aldrich or Alfa Aesar to assess the availability and cost of various precursors.

The pre-selection process used in our work can indeed be performed autonomously once the user provides a set of criteria that include: 1) which elements to consider in the target materials, 2) an upper limit on each material's energy above the convex hull, 3) the atmospheric conditions under which the materials will be synthesized, and 4) a threshold on the reaction energies that exist between each material and the gaseous species present in the specified atmosphere.

As we do not identify a clear trend in the success rate with respect to decomposition energy (**Fig. 2**), it is difficult to say whether the success rate would hold for a broader range of targets arising from loosened selection criteria.

The main text has been revised to include additional information on the pre-selection process. We have also created a GitHub repository with the scripts used for candidate filtering. Please find the corresponding details copied below.

Methods, Materials screening, Pages 16: The algorithm we used for identifying potential synthesis targets is available on GitHub (github.com/mattmcdermott/novel-materials-screening). It operates autonomously once given the following information: which elements to consider in the target materials, how large an upper limit to impose on each material's energy above the convex hull, the atmospheric conditions under which the materials will be synthesized, and a threshold on the reaction energies that exist between each material and the gaseous species present in the specified atmosphere. The algorithm then scrapes the Materials Project and produces a list of candidate materials that satisfy these criteria. Additional filtering may be considered based on the availability and cost of precursors for each target. While this is done manually in the current version of the algorithm, potential improvements could automate the process by leveraging online data from chemical inventory lists and vendor websites.

Code Availability, Page 20: The screening algorithm we used for identifying potential synthesis targets is available at github.com/mattmcdermott/novel-materials-screening.

Comment 5:

Can the authors give a prediction on what their results mean for the future? If the impressive success rates would hold true for other materials classes: Are we done with discoveries in a few years? Or what do you rate as limits? From the materials screening section it reads a bit like that there is not much more to discover as you screened all theoretical predictions in the Materials Project and only 146 feasible candidates were left from which you showed in 17 days what can be synthesized. I don't think that is the message which should be conveyed. The authors could consider adding some sentences where they explain the possibilities which are still open in the materials search space and put their approach in the right perspective.

Response:

We certainly did not want to give this impression. First of all, the Materials Project (MP) is far from complete in its coverage of "novel compounds" as it has never performed an exhaustive search of new (unreported) stable compounds across all chemical spaces. Novel compounds in MP most often arise serendipitously from research projects carried out in a particular area. Additionally, MP has very limited coverage of non-stoichiometric or disordered compounds. We therefore believe that many new and synthesizable compounds could be identified if MP was augmented using a more systematic approach, *e.g.*, by using machine learning algorithms trained to identify stable materials. So there certainly is much left to discover!

We would also like to point out that rather strict filtering criteria were used when identifying targets for these initial experiments in the A-Lab. We chose compounds that are stable or very close to the convex hull (< 10 meV/atom), have zero thermodynamic driving force to react with any gaseous species present in air throughout a wide range of temperatures (600-1100 °C), and contain just a subset of all possible elements, excluding those that are toxic, radioactive, or exceedingly rare. Were any of these requirements to be loosened, a much larger number of candidate materials would be produced.

We have modified the **Outlook** section to improve our messaging regarding future opportunities. These changes are copied below.

Outlook, Page 12: Our paper demonstrates that autonomous research agents can dramatically accelerate the pace of materials research. Researchers initialized the A-Lab by proposing 58 target materials, which were successfully realized at a rate of > 2 new materials per day with minimal human intervention. Such rapid discovery points to a vast landscape of opportunities in materials synthesis and development. While the current work focused on a limited subset of all possible synthesis targets, many new candidates await evaluation. As the breadth of ab-initio computations continues to grow, so will this list of novel materials.

Comment 6:

Any perspective on microstructure, which is also highly relevant for useful materials properties? In other words: it is not sufficient to “only” discover new materials, but their microstructure is also highly important.

Response:

We agree that microstructure plays an important role in optimizing the synthesis of materials and their properties. While this was not considered in our current project, we believe that it could be incorporated in later iterations of the A-Lab. To this end, we have installed a tabletop SEM and plan to integrate it into the lab’s workflow to provide characterization of each sample’s microstructure. After the integration of SEM, we anticipate that microstructure may be tailored by using standard optimization techniques to achieve the desired properties (*e.g.*, particle size and shape) with respect to several control handles including temperature, hold time, and heating/cooling rates.

The possibility of optimizing morphology is now mentioned in the **Outlook** (copied below).

Outlook, Page 12: The systematic nature of the A-Lab provides a unique opportunity to answer fundamental questions regarding the factors that govern the synthesizability of novel materials, serving as an experimental oracle to validate predictions made based on data-rich resources such as the Materials Project. In future iterations of the platform, such an oracle may be expanded to probe factors beyond synthesizability including microstructure and device performance.

Comment 7:

The concept of decomposition energies should be explained a bit more, earlier in the text. Also, the meaning of targets in this article should be directly explained in the main text when it first appears.

Response:

The manuscript has been revised such that the term “decomposition energy” is not introduced until it is accompanied by a definition. To further clarify its meaning, we have added a new figure to the SI that schematically illustrates the concept of decomposition energy. Please find it copied below.

Supplementary Fig. 2 | Schematic illustration of the decomposition energy. A binary convex hull is shown, where each circle represents a distinct phase that is thermodynamically stable. The y-axis represents the Gibbs free energy (G) while the x-axis represents a composition parameter. The decomposition energy (E_d) of a given target phase is determined by calculating the distance between its energy and the energy of a tieline formed by the two neighboring phases along the convex hull.

We have also modified the Introduction to elaborate on the meaning of “target materials.”

Introduction, Page 2: Given a set of **air-stable** target materials (*i.e.*, **desired synthesis products whose yield we aim to maximize**) screened using the Materials Project¹⁴, the A-Lab generates synthesis recipes using ML models trained on historical data from the literature, then executes these recipes with robotics.

Comment 8:

Where words such as paradigm, synergy, unique are used, it would be good to explain what they exactly mean in the context of the article.

Response:

To avoid ambiguity, we have removed all such terms and replaced them with clearer phrasing. The corresponding revisions are copied below.

Abstract, Page 1: The high success rate demonstrates the effectiveness of AI-driven platforms for autonomous materials discovery and motivates further integration of computations, historical knowledge, and robotics.

Introduction, Page 2: With its high success rate in validating predicted materials, the A-Lab showcases the collective power of ab-initio computations, ML algorithms, accumulated historical knowledge, and automation in experimental research.

Outlook, Page 12: The A-Lab demonstrates this by combining modern theory- and data-driven machine learning techniques with a modular workflow that can discover novel materials with minimal human input.

Comment 9:

The authors could consider explaining more in the main text on the data interpretation and decision-making abilities of the experimental agent.

Response:

While the data interpretation and decision making are both critical to the A-Lab's operation, their explanation in the main text is restricted by the word limit imposed by *Nature*. To aid the reader in understanding these methods while conserving space, the main text has now been revised to include clearer references to previous publications where each is described in detail. We have also added several notes to the SI with detailed walkthroughs of each method's application. Please find these notes copied in our response to Comment 1 from Reviewer #3.

Autonomous materials discovery platform, Page 4: If these literature-inspired recipes fail to produce > 50% yield for their desired targets, the A-Lab continues to experiment using ARROWS³, an active learning algorithm that integrates ab-initio computed reaction energies with observed synthesis outcomes to predict solid-state reaction pathways¹⁷.

Autonomous materials discovery platform, Page 5: The phase and weight fractions of the synthesis products are extracted from their XRD patterns by probabilistic ML models trained on experimental structures from the ICSD following methodology outlined in previous work^{18,19}.

Comment 10:

For multi-component compounds: Is pairwise reaction analysis (**Fig. 3**) sufficient?

Response:

Pairwise reaction analysis can in principle be applied to systems with an arbitrary number of components, so long as the corresponding phases react in pairs. Though, we do anticipate there to be more exceptions in systems containing many components. For example, multi-component oxides often form eutectic mixtures that melt at moderately low temperature, which would cause the associated reaction pathway to deviate from a simple pairwise sequence. It may also be more difficult to characterize reaction pathways when many precursors are used to synthesize multi-component materials, as their peaks are likely to overlap with one another in the corresponding XRD patterns.

We have added a note to the SI (copied below) that outlines these limitations.

Supplementary Note 12. Exceptions to pairwise reaction analysis

The active learning algorithm (ARROWS³) implemented in the A-Lab learns from synthesis outcomes by determining which pairwise reactions led to the formation of any newly observed products. From previous studies that utilized *in situ* XRD¹²⁻¹⁴, there is an abundance of evidence supporting the idea that solid phases generally react in pairs. Such reactions occur locally at the interfaces between precursors, where little diffusion is needed to form the resulting product. However, there are likely to be exceptions to this rule when precursors or reaction intermediates deviate from the solid state. For example, we anticipate that reactions may take place between more than two phases at a time when melting occurs. This would enable atoms to travel more freely throughout the sample, thereby alleviating the requirement

that two phases react locally at an interface. Indeed, recent work has shown that reactions proceed in a non-pairwise sequence when a multi-component oxide forms a eutectic melt¹⁵.

Pairwise reaction analysis can also be complicated by difficulties in characterization. For example, as more precursors are introduced to synthesize targets with many components, it may be challenging to interpret their XRD patterns owing to substantial peak overlap. Such cases may therefore warrant the use of additional characterization techniques that can spatially resolve distinct phases (*e.g.*, SEM/EDS measurements). We note that even in cases where pairwise reaction analysis is insufficient, or where the synthesis products are not completely characterized, the active learning algorithm will continue to operate. It will simply do so with reduced efficiency as it may not learn which precursors contribute to the formation of a desired synthesis product or its competing phases.

Comment 11:

Regarding **Fig. 4**, it is not clear what frequencies means here, cannot find it in the Figure.

Response:

We agree that the use of “frequency” was not clear in the original caption for **Fig. 4**. The wording has now been revised to state the purpose of the figure more clearly. These revisions are copied below.

From the caption of Fig. 4: The 17 target materials that could not be synthesized by the A-Lab, where each is categorized by the feature that complicates its synthesis. One target ($\text{Ta}_4\text{PbO}_{11}$) is excluded from this list, as it is metastable and therefore was predictably unobtained in favor of its stable competitors. The challenges in synthesizing the remaining 16 stable targets fall under two categories: experimental barriers (blue, 13 targets) and computational barriers (green, 3 targets). We distinguish these barriers into four distinct failure modes: slow reaction kinetics, precursor volatility, product amorphization, and limitations associated with DFT calculations performed at 0 K. A schematic explanation for each failure mode is provided.

Reviewer #3

General Assessment:

This paper presents an autonomous laboratory for the accelerated synthesis of novel inorganic materials, and demonstrate this with the synthesis of 41 novel compounds (out of a set of 58 targets).

The work is ambitious in its concept, aiming to go beyond previous autonomous materials experiments (*e.g.*, refs 3-6 in the MS) by folding in literature mining and computational predictions. To some degree, most of the individual components in the overall method are known but bringing all of them together in a successful experiment is non-trivial indeed, and outcome of the work (41 new compounds) is really impressive. I feel it is the kind of proof-of-concept of a new way of doing research that might fit in a journal such as this.

My main criticisms would be presentational - I found the manuscript a little difficult to review and quite hard to follow. This is, in part, because the authors are trying to pack a lot of concepts (automation, chemistry, algorithms, literature scraping) into a rather short format, but nonetheless, I genuinely doubt that researchers who are not close to this area will find the paper easy to digest, at least beyond the high-level concepts, aims, and outcomes (which are clear enough). This is exacerbated, I think, by the fact that the Supporting Information is not very long or detailed, as normalized to the complexity of what has been done. I have seen papers in this general area with far more technical detail in the ESI, and I think for such new methods this is essential. At times, it almost felt like the paper read as a summary or highlight, rather than a research paper. I don't think this is really the fault of the authors, who are excellent communicators, it is really just a symptom of attempting to explain a really complex and multifaceted study in rather little space. Length constraints aside, I think there are some ways that the paper might be restructured to address these issues.

Response:

We thank the reviewer for their feedback. We have made changes to the manuscript that we believe make it clearer and more understandable while keeping within the constraint of the Editor's request to shorten the main text. This includes significant additions to the SI and Extended Data. The changes suggested by the reviewer are addressed point-by-point below.

Comment 1:

Give at least one 'walked-through example' of how a single material was discovered. There are some details about specific compounds in the main text (e.g., $\text{CaFe}_2\text{P}_2\text{O}_9$, $\text{Mo}(\text{PO}_3)_5$, $\text{La}_5\text{Mn}_5\text{O}_{16}$), but they are short, and it rather ends up reading like a potpourri of asides. This is partly because the authors report so many new compounds, which is impressive, but it makes for a hard read if you really want to understand the methodology. This could be backed up by other detailed 'walk throughs' in the ESI - I wouldn't expect that for all compounds, but it would be really useful to flesh out the approach, and could include some 'failed' recipes, too, to highlight where things can go wrong.

Response:

Two detailed walkthroughs of the decision-making process have been added to the SI and are now referenced in the main text. This includes one example of a successful optimization and another example demonstrating how a search space can become exhausted. Please find the corresponding notes copied below.

Supplementary Note 6. Successful optimization of $\text{CaFe}_2\text{P}_2\text{O}_9$ synthesis

The A-Lab's experimental campaign targeting $\text{CaFe}_2\text{P}_2\text{O}_9$ began with four synthesis recipes that were suggested by the literature-inspired machine learning algorithms. These include the following precursor sets, each evaluated at the same temperature (1100 °C) so that one furnace could be shared between them:

Set A: CaCO_3 , Fe_2O_3 , $(\text{NH}_4)_2\text{HPO}_4$

Set B: CaO , Fe_2O_3 , $(\text{NH}_4)_2\text{HPO}_4$

Set C: $\text{Ca}(\text{OH})_2$, Fe_2O_3 , $(\text{NH}_4)_2\text{HPO}_4$

Set D: CaO , Fe_3O_4 , $\text{NH}_4\text{H}_2\text{PO}_4$

These produced glassy samples with low target yield and poor signal-to-noise ratio, likely due to melting at high temperature and subsequent amorphization upon cooling. As such, the same precursor sets were next evaluated at a lower temperature (800 °C). They were all found to produce identical phases: $\text{Ca}_3(\text{PO}_4)_2$ and FePO_4 . From these results, the active learning algorithm (ARROWS³) learned five pairwise reactions that occurred ≤ 800 °C, each listed below:

Using its interface with the Materials Project, the algorithm also determined that there is little thermodynamic driving force (8 meV/atom) remaining to form the target when $\text{Ca}_3(\text{PO}_4)_2$ and FePO_4 precede it. As such, it proposed two new sets of precursors to be tested at 800 °C:

Set E: CaO , Fe_2O_3 , $\text{NH}_4\text{H}_2\text{PO}_4$

Set F: CaO , Fe_3O_4 , $(\text{NH}_4)_2\text{HPO}_4$

In each set, ARROWS³ predicted a reaction between CaO and the phosphate precursor to form $\text{Ca}_3(\text{PO}_4)_2$ at a temperature ≤ 800 °C. However, the algorithm did not predict the formation of FePO_4 in either set as the below precursor pairs were not yet tested:

After evaluating both precursor sets at 800 °C, it was found that **Set E** followed a similar reaction pathway as **Sets A-D**. However, **Set F** was clearly distinguished by the formation of new peaks in its corresponding XRD pattern that could not be attributed to any known phases reported in the Materials Project or the ICSD.

Based on the first six experimental outcomes, ARROWS³ identified two different reaction pathways, one originating from **Sets A-E**, and another originating from **Set F**. As detailed in Supplementary Note 3, each unique reaction pathway only needs to be probed once throughout the full range of temperatures to determine whether it will successfully lead to the target's formation. Accordingly, the algorithm chose one precursor set from the first pathway (**Set E**) and another from the second (**Set F**) to be evaluated at higher temperature (900 °C). While the products from **Set E** exhibit little change from those observed at 800 °C, it was found that **Set F** produced a new phase, $\text{CaFe}_3\text{P}_3\text{O}_{13}$. This phase is computed to have a large driving force (77 meV/atom) to react with CaO and form the target. Indeed, heating **Set F** at 900 °C led to the formation of $\text{CaFe}_2\text{P}_2\text{O}_9$ with a yield of $\sim 70\%$. The experimental campaign was therefore halted as the results satisfied our stopping criterion of $> 50\%$ target yield.

Supplementary Note 5. Exhausting the search space for BaGdCrFeO_6 synthesis

The A-Lab's experimental campaign targeting BaGdCrFeO_6 began with four literature-inspired synthesis recipes that included the following precursor sets with a hold temperature of 1000 °C:

Set A: BaCO_3 , Cr_2O_3 , Fe_2O_3 , Gd_2O_3

Set B: BaCO_3 , Cr_2O_3 , Fe_3O_4 , Gd_2O_3

Set C: BaO , Cr_2O_3 , Fe_2O_3 , Gd_2O_3

Set D: BaO₂, Cr₂O₃, Fe₂O₃, Gd₂O₃

Sets A and **B** produced samples that predominantly contained BaCrO₄, in addition to minority amounts of GdCrO₃ and Fe₂O₃. In contrast, the samples resulting from **Sets C** and **D** contained substantially larger amounts of GdCrO₃ and only small amounts of BaCrO₄. These outcomes suggest some differences between the reaction pathway of each set. To further investigate these differences, the active learning algorithm (ARROWS³) implemented in the A-Lab proposed that all four precursor sets be tested at lower temperature (700 °C). The products obtained from **Sets A** and **B** contained BaCrO₄ in addition to Gd₂O₃ and Fe₂O₃. From these results, the algorithm gained information regarding two reactions:

In contrast, the samples produced by **Sets C** and **D** at 700 °C predominantly contained unreacted precursors, with just minority amounts of BaCrO₄. The formation of GdCrO₃, while evident at 1000 °C for each set, was not yet initiated at a lower temperature of 700 °C. Accordingly, ARROWS³ learned the following information from these outcomes:

Because the four initially tested precursor sets were found to react and form BaCrO₄ at the lower bound of the temperature range considered (700-1000 °C), ARROWS³ next suggested a set of precursors that was not yet observed to form such intermediates. This set also excludes Fe₃O₄ as the algorithm learned that it simply oxidizes to Fe₂O₃ at a temperature lower than 700 °C.

Set E: Ba(OH)₂, Cr₂O₃, Fe₂O₃, Gd₂O₃

However, when evaluated at 700 °C, this precursor combination was found to produce large amounts of BaCrO₄, appearing similar to the samples made by **Sets A** and **B**. The algorithm gleaned the following information from this outcome:

At this point, ARROWS³ identified two unique reaction pathways characterized by the amount of BaCrO₄ observed at low temperature: **Sets A**, **B**, and **E** formed BaCrO₄ in majority amounts at 700 °C while **Sets C** and **D** produced only minority amounts of that phase and instead contained a larger weight fraction of unreacted precursors. To further investigate these synthesis routes, the algorithm selected one representative set (**D** and **E**) from each pathway to evaluate at 800 °C. In either case, the products made at this temperature appeared similar to those obtained at 1000 °C. **Set D** (**E**) contained large (small)

amounts of BaCrO₄ and small (large) amounts of GdCrO₃. From these results, ARROWS³ refined the temperature range in which GdCrO₃ forms:

The algorithm also determined that there was no need to probe additional temperatures (900 °C) as the products formed at 800 °C appeared similar to those obtained at 1000 °C. At this point, all unique reaction pathways had been exhausted. As such, no more experiments were performed, and the synthesis was deemed “failed.”

Comment 2:

Provide more detail about the active learning parts. The experimental aspects are clear enough and figure 1 shows the workflow (powder dose → heat → XRD), but the recipe generation / choice aspects, notwithstanding Figure 2, were harder to follow.

Response:

To assist the reader in understanding the methods used to generate synthesis recipes, we have added an example to the SI (copied below). We have also included clearer references to previous work, where the methods are described in detail.

Supplementary Note 3. Literature-inspired synthesis recipe generation for MgNi(PO₃)₄

To demonstrate the methods used to generate initial synthesis recipes for novel targets, we provide a detailed walkthrough of the process for one target, MgNi(PO₃)₄. Our literature-inspired recommendation engine generates the first recipe by proposing the most common precursors for Mg, Ni, and P in reported solid-state synthesis experiments: MgO, NiO, and NH₄H₂PO₄. Additional recipes are then generated using a similarity-based strategy¹⁶, which operates under the assumption that highly similar target materials can be produced using the same (or related) precursors. By using the PrecursorSelector encoding model¹⁶, we calculated and ranked the similarity between MgNi(PO₃)₄ and each of the 28,598 target materials contained in our database of 33,343 solid-state synthesis procedures extracted from 24,304 publications. BaMg₂(PO₄)₂ was identified as the target that is most similar to MgNi(PO₃)₄ and which is also synthesized using precursors different from the most common ones. The precursors used to synthesize BaMg₂(PO₄)₂ include BaCO₃, MgO, and (NH₄)₂HPO₄. A masked precursor completion model¹⁶ was then used to exchange the necessary elements (Ni/Ba) to reach our current target,

MgNi(PO₃)₄. This algorithm identified NiO as the most likely replacement for BaCO₃, resulting in the following set of precursors: NiO, MgO, and (NH₄)₂HPO₄.

Following the process outlined above, three more recipes are generated by referring to three less similar target materials, Ca₈LuMg(PO₄)₇, Mg_{1.9}Ni_{0.1}TiO₄, and Y₂MoO₆. The corresponding synthesis recipes are listed in Supplementary Table 2. For each set of precursors, the synthesis temperature was predicted using a pre-trained XGBoost regressor¹⁷ based on the composition and thermodynamic properties of MgNi(PO₃)₄ and its precursors. Although the predicted temperature may vary for each recipe, we chose to use one fixed temperature for each target to maximize the possibility of batching multiple precursor sets in a single furnace. The actual synthesis temperature for MgNi(PO₃)₄ was calculated by averaging the five temperatures proposed across these recipes and rounded to the nearest hundred (900 °C).

Supplementary Table 3 | Five precursor sets generated for the synthesis of MgNi(PO₃)₄ by using a literature-inspired recommendation engine. The similarity between MgNi(PO₃)₄ and each reference target is evaluated using the PrecursorSelector encoding model¹⁶ and ranges from -1 to 1. The reference target does not apply to the precursor set with index of 0 because this precursor set is generated based on the most common precursors for Mg, Ni, and P as reported in the literature.

Index	Reference target	Similarity to MgNi(PO ₃) ₄	Precursors for reference target	Suggested precursors for MgNi(PO ₃) ₄
0	N/A	N/A	N/A	MgO, NiO, NH ₄ H ₂ PO ₄
1	BaMg ₂ (PO ₄) ₂	0.901	BaCO ₃ , MgO, (NH ₄) ₂ HPO ₄	MgO, NiO, (NH ₄) ₂ HPO ₄
2	Ca ₈ LuMg(PO ₄) ₇	0.864	CaCO ₃ , Lu ₂ O ₃ , MgCO ₃ , NH ₄ H ₂ PO ₄	MgCO ₃ , NiO, NH ₄ H ₂ PO ₄
3	Mg _{1.9} Ni _{0.1} TiO ₄	0.830	MgCO ₃ , NiO, TiO ₂	MgCO ₃ , NiO, (NH ₄) ₂ HPO ₄
4	Y ₂ MoO ₆	0.425	Y ₂ (C ₂ O ₄) ₃ , MoO ₃	MgCO ₃ , Ni(OH) ₂ , (NH ₄) ₂ HPO ₄

Comment 3:

While the overall experimental scheme is clear enough (Figure 1), some of the details are not. For example, the description of the automated synthesis and PXRD process, which is complicated, could be made clearer and augmented with explanatory figures in the ESI.

Response:

We have substantially expanded the SI and the Extended Data to include additional notes and figures detailing the experimental workflow used by the A-Lab. Also included are several videos that show the lab in action. Please find these changes copied below and attached with our resubmission.

Supplementary Note 11. Specifications of robots in the A-Lab

The A-Lab contains three 6-axis robot arms: one Mitsubishi RV7-FL and two Universal Robots (UR5e), each with force sensors and bespoke grippers (Extended Data Fig. 2a-c) designed to accommodate different types of consumables. Here we refer to the Mitsubishi robot as R1, the first UR5e robot as R2, and the second UR5e robot as R3. The Mitsubishi and UR5e robots can handle maximum payloads of 7 kg and 5 kg, respectively, excluding the tools and grippers mounted on them. The precision of each robot is within ± 0.02 and ± 0.03 mm, respectively. R1 uses two custom grippers to interact with the Quantos powder dosing heads, crucibles, and plastic vials/mixing pots (Supplementary Video 1). The grip is controlled using pneumatic actuators, regulated by pressure to achieve the specified force setting. R2 uses a Robotiq Hand-E gripper to handle crucibles and ceramic racks (Supplementary Video 2). R3 uses a Robotiq 2F-85 gripper to handle plastic vials, crucibles, acrylic discs, and XRD sample holders (Supplementary Video 3). Both UR5e robots use grippers with adjustable force, controlled by electric motors. This mechanism is convenient for programming delicate sample transfers, especially for strong but brittle alumina crucibles. R1 has a reach of 908 mm while R2 and R3 have a reach of 850 mm, before being extended by the grippers.

R1 and R3 are each mounted on stationary platforms, while R2 is mounted on a linear rail so that it can transfer samples from the precursor preparation station to the box furnaces. This linear rail was designed and installed in coordination with Olympus Controls. The rail uses a 4000 mm belt-driven HMRB15CCD0-4000-CD500K100 from Parker with a precision of ± 0.05 mm. Limit sensors are employed on both ends of the rail for recalibration (Extended Data Fig. 2d). The belt is driven by 3:1

PV60TA-003 (Parker) gearbox connected to LMDCE853C Novanta IMS Lexium stepper motor with ethernet connection that directly interfaces with UR teach pendant through URCap program provided by Schwarz Automation Inc. The aluminum platform to mount the linear rail is designed and manufactured by Olympus Controls. The sub-millimeter repeatability and direct power connection to the wall allows for little-to-no downtime required for maintenance and recalibration. While a linear rail only extends the work envelope in one axis, it also ensures improved repeatability relative to a system with increased degrees of freedom.

A carousel with four quadrants and a light gate is used in the precursor preparation station to arrange samples and place them for collection by the first UR5e robot (R2), as depicted in Extended Data Fig. 2e. For most operations, the carousel moves in coordination with the robot (R1) that prepares samples prior to heating. However, R2 can also request control when it needs to collect the samples and transfer them to the box furnaces. Once control is given, the carousel is directed to rotate such that the proper quadrant faces R2 and the samples are accessible from the robot's side (Supplementary Video 2). The use of this carousel ensures that many samples can be made and stored simultaneously in the precursor preparation station, allowing some queue to be developed prior to the heating step.

The videos described below can be opened by using the links provided. They are also attached as files in the Supplementary Material.

Supplementary Video 1: <https://www.youtube.com/watch?v=-XUUI-ThU1Y&feature=youtu.be>

Robot arm R1 (Mitsubishi) handling powders and slurries in the sample preparation station used to dispense and mix precursors prior to heating. The video is played at 20-times speed.

Supplementary Video 2: <https://youtu.be/26K5r68fzwQ>

Robot arm R2 (UR5e) moving crucibles from the sample preparation station to the box furnaces. The video is played at 20-times speed.

Supplementary Video 3: <https://www.youtube.com/watch?v=UILAUEkd06w>

Robot arm R3 (UR5e) retrieving powder samples (post-annealing) and cooperating with an Aeris X-ray diffractometer for their characterization. The video is played at 12-times speed.

Extended Data Fig. 2 | Robotic installations for sample transfer in the A-Lab. Grippers on the UR5e robotic arms that are used for **a**, sample preparation, **b**, loading/unloading of crucible racks to/from the box furnaces, and **c**, sample retrieval and characterization. **d**, Linear rail used to increase the working envelope of the robotic arm that loads/unloads crucible racks to/from the furnaces. **e**, Carousel used to organize and move samples in the sample preparation station.

Extended Data Fig. 3 | Communication protocols connecting each module in the A-lab. A local area network (LAN) is built to connect all the pieces of the A-Lab with a control computer using an RS-485 interface (or DB-25 for the box furnaces). Each module on the RS-485 interface has an IP assigned to enable communication with the computer. For enhanced cybersecurity, only the control computer has access to the internet while the LAN is isolated from it.

Reviewer Reports on the First Revision:

Referees' comments:

Referee #1 (Remarks to the Author):

I found that the authors have sufficiently addressed my comments and other reviewers, and I therefore recommend for publication of this manuscript in Nature.

Referee #2 (Remarks to the Author):

The authors addressed all my questions. I recommend publication as is.